# Prognostic Significance of Preoperative Inflammation Markers on the Long-Term Outcomes in Peritoneal Carcinomatosis from Ovarian Cancer

**DOI:** 10.3390/cancers16020254

**Published:** 2024-01-05

**Authors:** Irina Balescu, Mihai Eftimie, Sorin Petrea, Camelia Diaconu, Bogdan Gaspar, Lucian Pop, Valentin Varlas, Adrian Hasegan, Cristina Martac, Ciprian Bolca, Marilena Stoian, Cezar Stroescu, Anca Zgura, Nicolae Bacalbasa

**Affiliations:** 1“Carol Davila” University of Medicine and Pharmacy, 020021 Bucharest, Romania; 2Department of Surgery, “Carol Davila” University of Medicine and Pharmacy, 020021 Bucharest, Romania; mihaieftimie@yahoo.com (M.E.); sorinpetrea@yahoo.com (S.P.); bogdangaspar@yahoo.com (B.G.); nicolae_bacalbasa@yahoo.ro (N.B.); 3Department of Visceral Surgery, Center of Excellence in Translational Medicine “Fundeni” Clinical Institute, 022328 Bucharest, Romania; cezarstroescu@yahoo.com; 4Department of Surgery, “Ion Cantacuzino” Clinical Hospital, 010024 Bucharest, Romania; 5Department of Internal Medicine, “Floreasca” Clinical Emergency Hospital, 014461 Bucharest, Romania; cameliadiaconu@yahoo.com; 6Department of Internal Medicine, “Carol Davila” University of Medicine and Pharmacy, 020021 Bucharest, Romania; marilenastoian@yahoo.com; 7Department of Visceral Surgery, “Floreasca” Clinical Emergency Hospital, 014461 Bucharest, Romania; 8Department of Obstetrics and Gynecology, “Carol Davila” University of Medicine and Pharmacy, 020021 Bucharest, Romania; lucianpop@yahoo.com (L.P.); valentinvarlas@yahoo.com (V.V.); 9Department of Obstetrics and Gynecology, National Institute of Mother and Child Care, Alessandrescu-Rusescu, 127715 Bucharest, Romania; 10Department of Obstetrics and Gynecology, “Filantropia” Clinical Hospital, 011171 Bucharest, Romania; 11Department of Urology, Sibiu Emergency Hospital, Faculty of Medicine, University of Sibiu, 550245 Sibiu, Romania; adrianhasegan@yahoo.com; 12Department of Anesthesiology, Fundeni Clinical Hospital, 022328 Bucharest, Romania; cristina.martac@drd.umfcd.ro; 13Department of Thoracic Surgery, ‘Marius Nasta’ National Institute of Pneumology, 010024 Bucharest, Romania; ciprianbolca@yahoo.com; 14Faculty of Medicine and Health Sciences, Sherbrooke University, Sherbrooke, QC J1K 2R1, Canada; 15Department of Thoracic Surgery, ‘Charles LeMoyne’ Hospital, Longueuil, QC J4V 2H1, Canada; 16Department of Internal Medicine and Nephrology, Dr. Ion Cantacuzino Hospital, 010024 Bucharest, Romania; 17Department of Medical Oncology, Oncological Institute Prof. Dr. Al. Trestioreanu, 022328 Bucharest, Romania; medicanca@gmail.com; 18Department of Medical Oncology, “Carol Davila” University of Medicine and Pharmacy, 020021 Bucharest, Romania

**Keywords:** peritoneal carcinomatosis, preoperative inflammation, platelet-to-lymphocyte ratio, monocyte-to-lymphocyte ratio, systemic inflammation index

## Abstract

**Simple Summary:**

Ovarian cancer is still one of the deadliest malignancies affecting women worldwide which is associated with poor long-term outcomes even in certain cases in which complete debulking is achieved. In this respect, attention has been focused on identifying other prognostic factors which might indicate which cases are expected to have the best long-term outcomes such as the procoagulant or inflammatory status. The aim of the current paper is to analyse the correlation between the most frequently cited inflammatory and tumoral markers and the long-term outcomes of ovarian cancer patients.

**Abstract:**

Ovarian cancer remains one of the most lethal gynaecological malignancies affecting women worldwide; therefore, attention has been focused on identifying new prognostic factors which might help the clinician to select cases who could benefit most from surgery versus cases in which neoadjuvant systemic therapy followed by interval debulking surgery should be performed. The aim of the current paper is to identify whether preoperative inflammation could serve as a prognostic factor for advanced-stage ovarian cancer. Material and methods: The data of 57 patients who underwent to surgery for advanced-stage ovarian cancer between 2014 and 2020 at the Cantacuzino Clinical Hospital were retrospectively reviewed. The receiver operating characteristic curve was used to determine the optimal cut-off value of different inflammatory markers for the overall survival analysis. The analysed parameters were the preoperative level of CA125, monocyte-to-lymphocyte ratio (MLR), platelet-to-lymphocyte ratio (PLR), neutrophil-to-lymphocyte ratio (NLR) and systemic inflammation index (SII). Results: Baseline CA125 > 780 µ/mL, NLR ≥ 2.7, MLR > 0.25, PLR > 200 and a systemic immune inflammation index (SII, defined as platelet × neutrophil–lymphocyte ratio) ≥ 84,1000 were associated with significantly worse disease-free and overall survival in a univariate analysis. In a multivariate analysis, MLR and SII were significantly associated with higher values of overall survival (*p* < 0.0001 and *p* = 0.0124); meanwhile, preoperative values of CA125, PLR and MLR were not associated with the overall survival values (*p* = 0.5612, *p* = 0.6137 and *p* = 0.1982, respectively). In conclusion, patients presenting higher levels of MLR and SII preoperatively are expected to have a poorer outcome even if complete debulking surgery is performed and should be instead considered candidates for neoadjuvant systemic therapy followed by interval surgery.

## 1. Introduction

Although imaging techniques have significantly improved in the last decade, ovarian cancer still represents a very difficult diagnostic to establish especially in the early stages of the disease. The most cited marker for ovarian cancer diagnosis remains CA125, a molecule which exhibits increased values especially in advanced stages of the disease [1]; moreover, there are also cases in which ovarian cancer cells do not induce the apparition of increased levels of this antigen due to their incapacity to secrete it; in such cases, normal ranges of CA125 are encountered for a long period of time [2,3]. On the other hand, we should not omit the fact that a significant number of patients present increased CA125 levels in the absence of malignant ovarian tumours, a higher amount of this antigen being also caused by ovarian and uterine benign conditions, pregnancy, autoimmune disorders, liver or cardiac disfunctions or non-gynaecological malignancies [4,5,6]. Therefore, attention has been focused on identifying other laboratory tests which might orientate the diagnostic and provide more specific information regarding the overall prognostic in ovarian cancer patients.

In the last decade, a particular interest has been given to the study of the possible correlation between oncogenesis and inflammation, and multiple studies have come to demonstrate that the presence of chronic inflammation creates a favourable microenvironment for tumorigenesis. Therefore, it seems that the presence of a higher number of platelets, neutrophils and monocytes increase the number of proinflammatory cytokines and tumour growth factors, while a higher number of lymphocytes has, as expected, a protective role against tumoral development and spread. In this respect, multiple combinations have been proposed such as platelet-to-lymphocyte ratio, lymphocyte-to-monocyte ratio, neutrophil-to-lymphocyte ratio or systemic inflammatory indexes [7,8,9]. The purpose of the current paper was to investigate the possible correlation between preoperative inflammation markers and long-term outcomes in advanced-stage ovarian cancer submitted to surgery as the first therapeutic option.

## 2. Materials and Methods

After receiving the approval of the Ethics Committee of Ion Cantacuzino Clinical Hospital (no. 156/2023), the data of patients who underwent surgery for advanced-stage ovarian cancer between 2014 and 2020 were retrospectively reviewed. All data regarding preoperative, intraoperative and early postoperative outcomes were obtained from patients’ medical records, while data regarding survival were obtained from the National Population Register.

Inclusion criteria were represented by a histopathological confirmation of advanced-stage epithelial ovarian cancer (FIGO stages IIIC and IV), surgery with curative intent as the first therapeutic strategy and an age over 18 years. Exclusion criteria were represented by an age under 18 years, nonepithelial ovarian carcinomas, the administration of systemic neoadjuvant chemotherapy, a stage lower than IIIC at the histopathological findings, a history of other neoplastic diseases, the presence of autoimmune or haematological disorders and the confirmation of severe infections which might influence the preoperative values of platelets and white blood cell lines. A total number of 57 patients were considered to be eligible for this study. In all cases, a preoperative complete blood count was retrieved 24 h before surgery and was processed by using a MINDRAY BC-5380 3DIFF analyser; meanwhile, data regarding age, preoperative level of CA125, albumin and C-reactive protein were retrospectively reviewed. The CA125 level was determined by using the RayBio Human CA125 ELISA Kit. In all cases the aim of the surgical procedure was to achieve complete debulking surgery defined as no visible residual disease; therefore, resection was defined as R0 if no residual disease was achieved, R1 if the residual disease with a diameter lower than 1 cm was present at the end of the surgical procedure, and R2 if the residual disease was larger than 1 cm. The early postoperative complications were classified according to the Dindo–Clavien scale, while the long-term outcomes were evaluated according to progression-free and overall survival rate. Progression-free survival was defined as the interval between the initial surgical procedure and the diagnostic of relapse, while overall survival was defined as the interval between initial surgery and time of death. A follow-up was performed in all cases at three months during the first two years and six months until the five-year mark and consisted of chest computed tomography, abdominal and pelvic magnetic resonance imaging and tumoral markers—CA125 dosage. Chi-square and Student’s *t*-tests were used in order to compare categorical and continuous variables, respectively.

Univariate and multivariate analyses were performed by using the Cox proportional hazard model and the multivariate Cox proportional hazard model, respectively, where in the multivariate analysis, we introduced the factors showing statistical significance in the univariate analysis. In all cases, a *p*-value lower than 0.05 was considered as statistically significant. The median overall survival curves were obtained by performing a Kaplan–Meyer analysis and were compared using the log-rank test. All data analyses were performed by using IBM SPSS statistical software version 18.0 (SPSS Inc., Chicago, IL, USA). The receiver operating characteristic (ROC) curve was used with the Youden index (maximum (sensitivity + specificity − 1)) in order to determine the cut-off values for different variables in order to predict the possibility of achieving an R0 resection. Therefore, the main parameters were represented by the systemic inflammatory index SII, defined as the platelet × neutrophil–lymphocyte ratio with an area under the ROC curve (AUC) of 0.787 (sensibility = 0.83, 1 − specificity = 0.29), a neutrophil-to-lymphocyte ratio with an AUC of 0.779 (sensibility = 0.85, 1 − specificity = 0.29), a monocyte-to-lymphocyte ratio with an AUC OF 0.776 (sensibility = 0.82, 1 − specificity = 0.29), a platelet-to-lymphocyte ratio, with an AUC of 0.700 (sensibility = 0.725, 1 − specificity = 0.353) and CA125, with an AUC of 0.700 (sensibility = 0.675, 1 − specificity = 0.294) (Figure 1). By using the ROC curve, the cut-off values were established at 841,000 for the platelet × neutrophil–lymphocyte ratio (SII), 2.7 for the neutrophil-to-lymphocyte ratio, 0.25 for the monocyte-to-lymphocyte ratio, 200 for the platelet-to-lymphocyte ratio and 780 for CA125.

## 3. Results

A total number of 57 patients were considered as eligible for this study, the mean age at the time of initial diagnosis being 56 years old (range 25–83 years old); 31 were younger than 60 years of age, while the other 26 cases were older than 60 years of age. Among the 26 cases, there were 17 cases with associated comorbidities while among patients younger than 60 years, comorbidities were present in 5 cases (*p* < 0.0001). Comorbidities were represented by arterial hypertension in twelve cases, chronic hepatitis in four cases, obstructive pulmonary disease in three cases, diabetes mellitus in three cases. In all cases, surgery was the first-intent treatment and consisted of debulking surgery to no residual disease; however, this outcome was obtained in 28 patients younger than 60 years of age and in 18 cases among elderly patients (*p* = 0.08). Postoperative complications occurred in eight cases among elderly patients and in eight cases among younger patients (*p* = 0.771); however, severe complications imposing reoperation were encountered in a single case among patients younger than 60 years old and in five cases among elderly patients. Moreover, three out of the five patients with severe complications died among the first postoperative month, while among younger cases, no death occurred within the first month postoperatively. The clinicopathological characteristics of the patients are presented in Table 1.

When it comes to the preoperative laboratory tests, the mean CA125 level at the time of diagnosis was 2892 U/mL (range 78–14,535 U/mL), the mean level of serum albumin was 3.25 g/dL (range 1.6–5.1 g/dL), while the mean level of total proteins was 6.03 g/dL (range 3.1–8.3 g/dL). When analysing the preoperative complete blood count, the mean values were 11.88 g/dL for haemoglobin (range 7.8–16.1 g/dL), 367,624/µL for circulating platelets (range 167,000–848,100/µL), 5176/µL for neutrophils (range 1475–12,980/µL), 1471/µL for lymphocytes (range 472–3120/µL) and 643/µL for monocytes (range 138–1520/µL). Using these values, we went further and calculated the systemic inflammatory index (neutrophils × platelets/lymphocytes), and a mean value of 1,757,495 was obtained (range 204,265–7,548,167).

Based on the cut-off values obtained by using the ROC curve, we further stratified patients among two groups: 20 patients with an SII lower than 841,000 and 37 cases with an SII higher than 841,000, 17 patients a neutrophil-to-lymphocyte ratio lower than 2.7 and 40 cases with a higher value of this parameter, 21 cases with a monocyte-to-lymphocyte ratio lower than 0.25 and 36 cases with a monocyte-to-lymphocyte ratio (MLR) higher than 0.25, 23 cases with a platelet-to-lymphocyte ratio (PLR) lower than 200 and 34 cases with a value higher than 200, and 25 cases with a CA125 value lower than 780 U/L and 32 cases with a value higher than 780 U/L.

The preoperative and intraoperative features of the study group according to the level of CA125, MLR, NLR, PLR and SII are presented in Table 2.

As it can be observed, there was no significant difference between the preoperative values of CA125, MLR, NLR, PLR and SII among patients younger or older than 60 years old. When it comes to the histology of the tumours, serous lesions seemed to be associated with higher values of CA125 and with higher values of SII, while the other parameters were similar between the different histopathologic subgroups. However, after applying the Bonferroni correction, there was no significant correlation between preoperative values of CA125 or SII and the different histopathological subtypes (all obtained Bonferroni *p*-values being higher than 0.05). Interestingly, the degree of differentiation did not significantly influence the values of preoperative CA125 and the degree of preoperative inflammation. In order to establish a possible correlation between these parameters and the extent of the disease, we analysed the distribution of the ascites volume among the different subgroups as well as the peritoneal carcinomatosis index (PCI), and we observed that cases presenting a lower value of CA125 as well as lower values of MLR, NLR, PLR and SII reported significantly lower volumes of ascites as well as a lower value of the peritoneal carcinomatosis index. Moreover, in these cases, after applying the Bonferroni correction, the resulting differences remained statistically significant. As expected, patients presenting a more extended neoplastic process (defined by a higher value of CA125, a higher volume of ascites and higher PCI values presented significantly lower preoperative haemoglobin levels and higher rates of postoperative complications; interestingly, the rates of associated comorbidities were similar between the different groups. Although the presence of more extended neoplastic processes is expected to be associated with a higher rate of malnutrition defined by a lower albumin level, a statistically significant correlation was only observed between preoperative levels of CA125, NLR and PLR and lower albumin levels. When analysing the corelation between the completeness of the cytoreduction and preoperative CA125, MLR, NLR, PLR and SII levels, statistically significant differences were observed in all cases; therefore, patients with higher CA125, MLR, NLR, PLR and SII levels were more likely to achieve an incomplete cytoreductive surgical procedure (*p* = 0.001, *p* = 0.004, *p* = 0.04, *p* = 0.03 and *p* = 0.005, respectively). This fact was once again in concordance with the correlations regarding the extent of the disease, the presence of higher volumes of ascites and PCI being also associated with higher levels of CA125 and with a higher inflammatory condition, as mentioned before. However, when investigating the possible correlation between the BRCA status and inflammatory status, no statistically significant correlation could be established.

When it comes to the long-term outcomes, all investigated parameters proved to be significantly associated with the overall survival: patients with higher values of CA125 reported a mean overall survival of 16.97 months, while cases with lower CA125 values reported a mean overall survival of 43.61 months, *p* < 0.001; patients with higher values of MLR reported a mean overall survival of 15.62 months, while cases with lower MLR values reported a mean overall survival of 45 months, *p* < 0.001; patients with higher values of NLR reported a mean overall survival of 15.32 months, while cases with lower NLR values reported a mean overall survival of 44.2 months, *p* < 0.001; patients with higher values of PLR reported a mean overall survival of 17 months, while cases with lower PLR values reported a mean overall survival of 43.6 months, *p* < 0.001; and patients with higher values of SII reported a mean overall survival of 15 months versus 44 months for cases with lower SII values, *p* < 0.001 (Figure 2, Figure 3, Figure 4, Figure 5 and Figure 6).

A similar correlation was observed when studying the impact of the investigated parameters on the disease-free survival rate; therefore, when calculating the median overall survival for the entire group, a value of 14 months was obtained. However, patients with lower CA125 values reported a median disease-free survival of 28 months, while in cases with higher CA125 values, this value dropped off at 11 months (*p* = 0.003); patients with lower MLR values reported a median disease-free survival rate of 30 months, while cases with higher MLR values reported a median disease-free survival rate of 12 months (*p* = 0.002); patients with lower NLR values reported a median disease-free survival rate of 33 months, significantly higher than those with higher NLR values (in this group, that value was 13 months, *p* = 0.001); patients with a lower PLR reported a median disease-free survival rate of 28 months, significantly higher than those with higher PLR values (in this group, that value was 14 months, *p* = 0.01); and patients with a lower SII reported a median disease-free survival rate of 31 months, significantly higher than those with higher SII values (in this group, that value was 12 months, *p* = 0.006) (Figure 7, Figure 8, Figure 9, Figure 10 and Figure 11).

Meanwhile, when studying the possible correlation between BRCA status, inflammatory markers and disease-free and overall survival, we observed that BRCA-mutated patients with SII ≥ 841,000 and those with higher NLR values had significantly worse DFS compared to patients with SII  <  841,000 and with lower NLR  values. However, when studying the correlation between these parameters and overall survival, no significant correlation could be demonstrated.

We also performed a univariate analysis of prognostic factors influencing the overall survival, which demonstrated the patient’s age, PCI, CA125, PLR, MLR, NLR and SII values as the prognosis-influencing parameters. A multivariate analysis was then performed considering these parameters. In the multivariate analysis, preoperative MLR and SII were significantly associated with higher values of overall survival (*p* < 0.0001 and *p* = 0.0124); meanwhile, preoperative values of CA125, PLR and NLR were not associated with the values of overall survival (*p* = 0.5612, *p* = 0.6137 and *p* = 0.1982, respectively).

## 4. Discussion

In the current paper, we aimed to analyse the possible correlations between preoperative levels of CA125, preoperative systemic inflammation, the extent of the disease and long-term outcomes of advanced-stage ovarian cancer. As mentioned before, the levels of CA125 cannot be considered as a gold standard in order to identify ovarian cancer patients and to predict long-term outcomes [4,5,6]; therefore, attention was focused on identifying other prognostic markers which might provide an adequate identification of cases who could benefit most from debulking surgery followed by adjuvant chemotherapy or, oppositely, who could better benefit from neoadjuvant therapy followed be interval debulking surgery [10,11,12]. One of the most commonly cited prognostic factors which has been recently investigated in patients suffering from malignant diseases is represented by the inflammatory status of each patient. Therefore, it has been widely demonstrated that molecular and genetic mechanisms of tumorigenesis as well as tumour progression and angiogenesis are strongly influenced by the immune status of the host as well as by different homeostatic mechanisms [13,14,15,16,17]. In this respect, it has been demonstrated that platelets seem to play a central role in promoting tumoral cells’ growth, dissemination and colonization determining the apparition of metastatic deposits, especially through the process of degranulation and the release of proinflammatory cytokines (such as interleukin 6) and tumour growth factors (such as tumour growth factor β-TGF β). Meanwhile, the presence of a higher number of neutrophils is usually the sign of a higher tumoral burden and in turn is responsible for the secretion of cancer-promoting cytokines, which further provide an adequate tumoral environment for tumoral spread and seeding. As one of the most important inhibitors of tumoral dissemination, the lymphocytes also seem to play a crucial role in activating the immune response of the host against tumoral cells and in blocking tumoral dissemination and seeding. As demonstrated so far, in cancer patients a lower level of lymphocytes is to be expected, and therefore, the response of the host against malignant cells remains poor [16,17,18,19]. Moreover, circulating platelets seem to promote the development of the epithelial-to-mesenchymal transition while lymphocytes express antitumour activities by recognizing foreign antigens which are usually expressed at the surface of malignant cells [17,18,19,20].

Once these aspects had been widely demonstrated by immunological studies, attention was focused on investigating if the pretreatment values of these parameters (alone or in combination) might have a predictive value of the long-term outcomes of ovarian cancer patients; the reported results proved to be rather interesting, with significant results being reported so far. In this respect, investigators worldwide came to demonstrate that a higher number of platelets associated with a higher number of circulating neutrophils and monocytes and a lower level of lymphocytes significantly influenced the long-term outcomes of ovarian cancer. Moreover, this correlation proved to be present even when studying the influence of certain ratios such as the platelet-to-lymphocyte ratio, lymphocyte-to-monocyte ratio, neutrophil-to-lymphocyte ratio [20,21,22,23,24,25].

In ovarian cancer, a significant number of studies conducted at the beginning of the XXI century demonstrated that preoperative values of PLR or NLR represented significant prognostic factors for predicting the long-term outcomes of advanced-stage ovarian cancer; therefore, in the study conducted by Miao et al., the authors underlined the fact that preoperative values of NLR lower than 3.02 or of PLR lower than 207 are significant predictors for an increased long-term survival; therefore, the authors concluded that such patients would rather benefit from per primam debulking surgery rather than neoadjuvant chemotherapy followed by interval debulking [26]. However, more recent studies underlined that PLR had no such significant power of predicting long-term outcomes, a higher value of this parameter not being associated with a poorer disease-free and overall survival [27]. Therefore, attention was focused on identifying other prognostic markers which could integrate both the coagulant and the inflammatory status of the host. In this respect, a novel indicator has been imagined, the systemic inflammation index, SII, defined as the ratio between the platelets and neutrophils reported to lymphocytes (platelets × neutrophils/lymphocytes). SII proved to be an efficient prognostic marker in order to identify tumours with a poorer biological behaviour among colon, liver or renal carcinomas; moreover, it seems that a higher preoperative level of SII is associated with a higher number of cancer-circulating cells in patients with hepatocarcinoma, therefore predicting the risk of developing hematogenous metastases [21]. Moreover, Wang et al. underlined the fact that dynamic modifications of SII at six weeks postoperatively in patients with hepatocarcinoma had a predictive value in identifying cases with improved prognosis after resection [28]. Meanwhile, it seems that this parameter is useful for estimating the extent of the disease in peritoneal carcinomatosis from ovarian or colorectal origin [20,21,22,23,24,25] as well as the therapeutic response at chemotherapy alone or in association with monoclonal antibodies. Therefore, a recent study conducted on patients with peritoneal carcinomatosis from ovarian cancer demonstrated that patients presenting a pretreatment value of SII over 730 did not benefit after adjusting bevacizumab to standard chemotherapy alone while cases presenting a lower value of this parameter reported a significantly improved outcome after bevacizumab association [29].

One of the largest studies which was focused on the predictive value of SII on the long-term outcomes of advanced-stage ovarian cancer patients was conducted by Nie et al. and published in the journal *Gynecologic Oncology* in 2018; the study included 553 patients and demonstrated that cases presenting a preoperative value of SII higher than 612 were usually diagnosed with more advanced stages of the disease, presented more frequently with lymph node metastases and reported a lower progression-free survival interval. Meanwhile, such cases presented both in univariate and multivariate analyses a shorter disease-free and overall survival, demonstrating that this parameter should be considered as an independent prognostic factor for epithelial ovarian cancer patients [30].

A recent study which investigated the possible correlation between SII and the perioperative outcomes in advanced-stage ovarian cancer was published by Ramon-Rodriguez et al. in 2022; the study included 68 patients diagnosed with advanced-stage ovarian cancer who were submitted to debulking surgery and intraperitoneal chemo hyperthermia. Similar to our study, a significant correlation between the histopathological subtype and the intensity of the systemic inflammatory response could not be proved; however, a significant correlation between the values of these parameters and the disease-free and overall survival was demonstrated. When it comes to the prognostic factors impacting the long-term outcomes, Ramon-Rodriguez et al. demonstrated that PCI, SII values and the response to chemotherapy were the most important prognostic factors in both univariate and multivariate analyses [31].

One of the largest meta-analyses conducted on the issue of inflammatory status and overall prognosis in ovarian cancer was published in 2018 by Zhu et al. and demonstrated that the preoperative values of NLR and PLR represent valuable tools in order to identify patients with poorer long-term outcomes; moreover, the authors underlined the fact that the higher the cut-off values for NLR and PLR, the stronger the expected predictive effect [32]. More recently, an Italian study group conducted by Farolfi et al. demonstrated that NLR and SII also had a prognostic value at the time of relapse, lower values of NLR being considered as an independent predictive factor for platinum sensitivity in cases in which bevacizumab had not been previously administrated [33].

Moreover, systemic inflammatory markers have recently been investigated in the setting of early-stage ovarian cancer; therefore, cases with FIGO stage I ovarian cancer presenting higher preoperative values of PLR, SII and NLR seem to have poorer disease-free and cancer-specific survival rates. In this respect, a more complex postoperative therapy might be proposed in such cases in order to improve long-term outcomes [34]. Moreover, this correlation seems to be stronger in BRCA-mutated patients and apparently in early stages of the disease, this observation being particularly helpful in order to design a more personalized and efficient therapy, even in cases diagnosed with early-stage ovarian cancer [35].

Another important issue which should be discussed is the one regarding the possible correlation between BRCA status, inflammatory status and long-term outcomes. Therefore, in our study we demonstrated that BRCA-mutated patients with higher SII values and those with higher NLR values had significantly worse DFS compared to those with a lower SII and lower NLR values. However, this correlation failed to be demonstrated in regard to the overall survival. Similar results were published in 2021 by Marchetti et al., which underlined the fact that NLR and BRCA status significantly influenced the long-term outcomes in high-grade serous advanced-stage ovarian cancer. This information is particularly important in order to demonstrate that BRCA-mutant cases do not represent a single category of patients and therefore anti Poly (ADP-ribose) polymerase (PARP) inhibitors (known to provide the most remarkable response in BRCA-mutated patients) should not be routinely administrated in all cases presenting BRCA mutations [36].

When it comes to the strengths and limitations of this study, we consider that the main strength is related to the fact that all surgeries were performed by the same surgeon in the same hospital and mainly respected the same protocol; moreover, the specimens were analysed by the same histopathological team. Therefore, the “human” factor was almost completely eliminated. However, the main limitations of our study are represented by the relatively low number of patients and by the retrospective character of the study; larger, even multicentric, prospective studies are still needed in order to validate these results. Meanwhile, including cases with nonserous histology with different biological behaviours might also influence the results.

## 5. Conclusions

The preoperative inflammatory status defined through multiple parameters (MLR, PLR, NLR, SII), together with the preoperative values of CA125, seems to offer significant information regarding the extent of the disease in advanced-stage ovarian cancer and might provide a better selection of cases who could benefit most after per primam cytoreductive surgery versus cases in which neoadjuvant systemic therapy followed by interval debulking surgery should be performed. According to our findings, the best predictive value is given by MLR and SII, a higher preoperative value of these parameters being the sign of a more aggressive biology of the tumour and of the presence of more extended lesions, imposing the association of neoadjuvant chemotherapy in order to maximize the survival benefit.

## Figures and Tables

**Figure 1 cancers-16-00254-f001:**
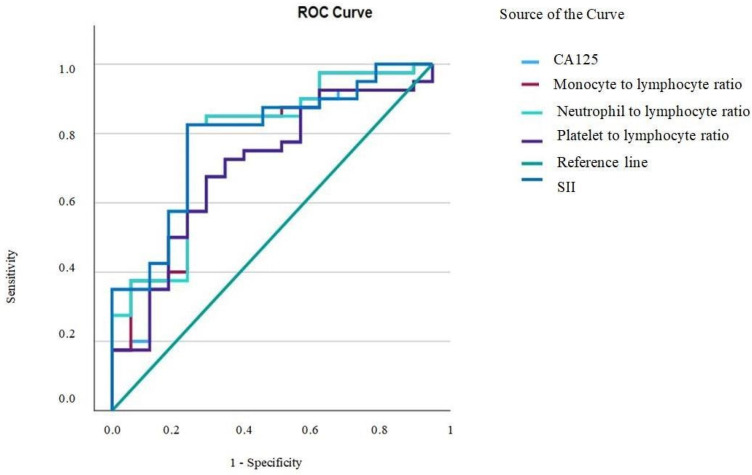
Receiver operating characteristics curve analysis of CA125, monocyte-to-lymphocyte ratio (MLR), neutrophil-to-lymphocyte ratio (NLR), platelet-to-lymphocyte ratio (PLR) and systemic immune inflammation index (SII).

**Figure 2 cancers-16-00254-f002:**
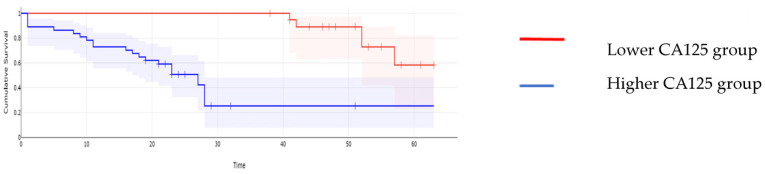
Kaplan–Meyer curves based on preoperative values of CA125: patients with higher values of CA125 reported a mean overall survival of 16.97 months, while cases with lower CA125 values reported a mean overall survival of 43.61 months, *p* < 0.001.

**Figure 3 cancers-16-00254-f003:**
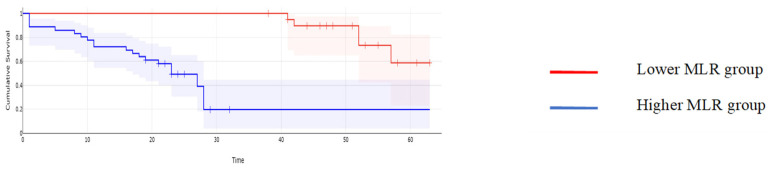
Kaplan–Meyer curves based on preoperative values of MLR: patients with higher values of MLR reported a mean overall survival of 15.62 months while cases, with lower MLR values reported a mean overall survival of 45 months, *p* < 0.001.

**Figure 4 cancers-16-00254-f004:**
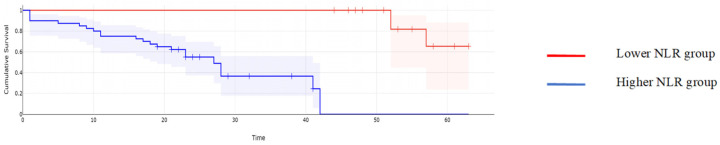
Kaplan–Meyer curves based on preoperative values of NLR: patients with higher values of NLR reported a mean overall survival of 15.32 months, while cases with lower NLR values reported a mean overall survival of 44.2 months, *p* < 0.001.

**Figure 5 cancers-16-00254-f005:**
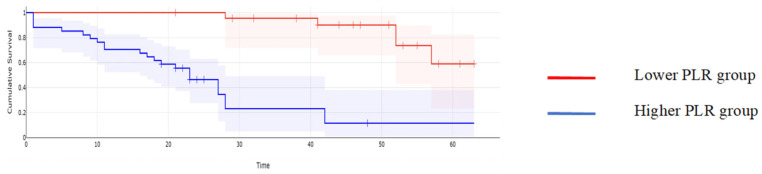
Kaplan–Meyer curves based on preoperative values of PLR: patients with higher values of PLR reported a mean overall survival of 17 months, while cases with lower PLR values reported a mean overall survival of 43.6 months, *p* < 0.001.

**Figure 6 cancers-16-00254-f006:**
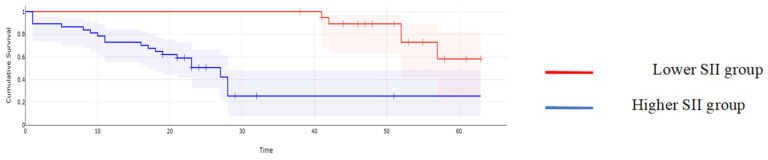
Kaplan–Meyer curves based on preoperative values of SII: patients with higher values of SII reported a mean overall survival of 15 months, while cases with lower SII values reported a mean overall survival of 44 months, *p* < 0.001.

**Figure 7 cancers-16-00254-f007:**
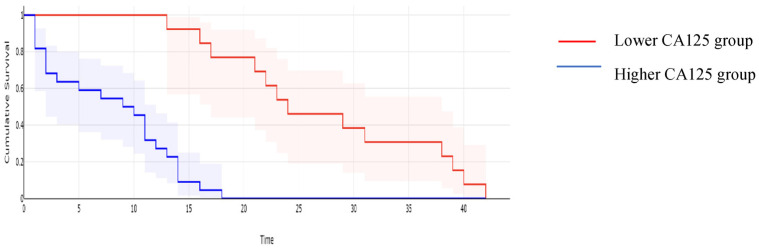
Kaplan–Meyer curves of DFS based on the preoperative levels of CA125. Patients with higher values of CA125 reported a median DFS rate of 11 months, while cases with lower CA125 values reported a mean overall survival of 30 months, *p* = 0.002.

**Figure 8 cancers-16-00254-f008:**
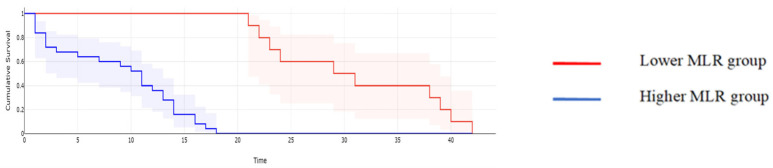
Kaplan–Meyer curves of DFS based on the preoperative levels of MLR. Patients with higher values of MLR reported a median DFS rate of 12 months, while cases with lower MLR 125 values reported a mean overall survival of 28 months, *p* = 0.003.

**Figure 9 cancers-16-00254-f009:**
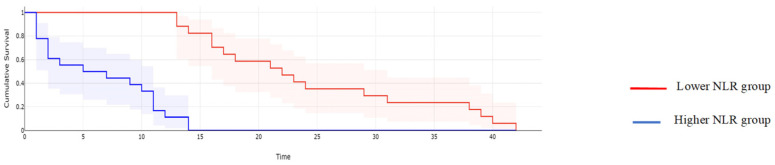
Kaplan–Meyer curves of DFS based on the preoperative levels of NLR. Patients with higher values of NLR reported a median DFS rate of 13 months, while cases with lower NLR values reported a mean overall survival of 33 months, *p* = 0.0001.

**Figure 10 cancers-16-00254-f010:**
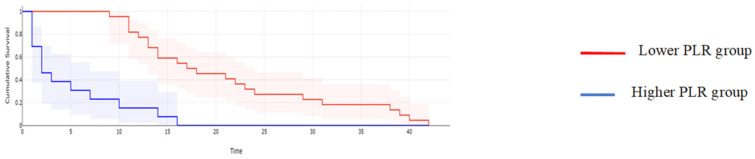
Kaplan–Meyer curves of DFS based on the preoperative levels of PLR. Patients with higher values of PLR reported a median DFS rate of 14 months, while cases with lower PLR values reported a mean overall survival of 28 months, *p* = 0.01.

**Figure 11 cancers-16-00254-f011:**
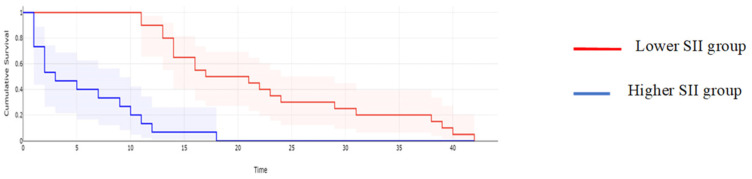
Kaplan–Meyer curves of DFS based on the preoperative levels of SII. Patients with higher values of SII reported a median DFS rate of 12 months, while cases with lower SII values reported a mean overall survival of 31 months, *p* = 0.006.

**Table 1 cancers-16-00254-t001:** Preoperative clinicopathological characteristics of the study group.

Characteristics	Value
Total number of cases	57
Mean age at the time of initial diagnosis	56 years old (range 25–83 years old)
Comorbidities	22 cases
Final FIGO stage:	
- IIIC - IV	51 6
Histology:	
- Serous - Endometroid - Mucinous - Clear cell - Other histology (carcinosarcoma in two cases, dysgerminoma in two cases and immature teratoma in one case)	35 5 5 7 5
Differentiation degree:	
- G1 - G2 - G3	6 15 36
BRCA status:	
- no mutation - BRCA1 mutation - BRCA2 mutation - BRCA 1-2 mutation	46 8 2 1
CA125 level:	
- <780 - >780	25 32
MLR level:	
- <0.25 - >0.25	21 36
NLR level:	
- <2.7 - >2.7	17 40
PLR level:	
- <200 - >200	23 34
SII level:	
- <841,000 - >841,000	20 37

**Table 2 cancers-16-00254-t002:** Distribution of patients based on the cut off levels of CA125, MLR, NLR, PLR and SII.

No. of Cases	CA125 (U/mL)			MLR			NLR			PLR			SII		
	<780	>780	*p*	<0.25	>0.25	*p*	<2.7	>2.7	*p*	<200	>200	*p*	<841,000	>841,000	*p*
Age			0.59			0.78			0.77			0.58			0.9
<60	15	16		12	19		10	21		14	17		11	20	
>60	10	16		9	17		7	19		9	17		9	17	
HP:			0.04			0.56			0.34			0.33			0.03
SE	18	17		15	20		14	21		23	12		19	16	
MU	2	3		1	4		3	2		3	2		4	1	
EN	2	3		3	2		2	3		2	3		3	2	
ClC	2	5		4	3		2	5		2	5		5	2	
OT*	1	4		2	3		1	4		2	3		2	3	
DD:			0.41			0.66			0.12			0.44			0.11
G1	4	2		4	2		4	2		4	2		3	3	
G2	10	5		8	7		6	9		9	6		8	7	
G3	11	25		9	27		7	29		10	26		9	27	
PCI:			0.002			0.001			0.003			0.01			0.001
<10	13	1		13	1		13	1		13	1		10	2	
10–15	11	10		8	23		3	17		4	13		8	15	
>15	1	21		1	21		1	21		6	20		2	20	
Ascites (mL) (mean)	1516	3003	0.002	1419	2894	0.03	1352	2775	0.009	1558	3026	0.01	1355	2889	0.008
Haemoglobin levels (g/dL):	12.5	11.3	0.008	12.6	11.3	0.005	12.9	11.4	0.002	12.5	11.3	0.009	13.1	11.4	0.005
Albumin (g/dL)			0.008			0.53			0.002			0.006			0.74
<3.5	8	22		4	10		6	26		7	24		4	10	
>3.5	17	10		17	26		15	10		16	10		16	27	
BRCA status:			0.217			0.15			0.09			0.32			0.19
Wild-type	22	24		19	27		16	30		20	26		18	28	
Mutated	3	8		2	9		1	10		3	8		2	9	
Comorbidities			1			0.77			0.77			1			0.41
Yes	10	12		9	13		8	16		10	14		10	14	
No	15	20		12	23		9	24		13	20		10	23	
CoC:			0.001			0.004			0.04			0.03			0.005
R0	25	21		21	25		16	30		22	24		20	26	
>R0	0	11		0	11		1	10		1	10		0	11	
Postop. complications:			0.02			0.03			0.02			0.008			0.03
yes	3	13		2	14		1	15		2	14		2	14	
no	22	19		19	22		16	25		21	20		18	23	
Total	25	32	-	21	36	-	17	40	-	23	34	-	20	37	-

HP—histopathological findings; SE—serous; MU—mucinous; EN—endometroid; ClC—clear cell; OT—other; DD—differentiation degree; PCI—peritoneal carcinomatosis index; CoC—completeness of cytoreduction; Postop.—postoperative. OT*—carcinosarcoma in two cases, dysgerminoma in two cases and immature teratoma in one case.

## Data Availability

Data are available upon reasonable request.

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
