# Peer review of "Prognostic Significance of Preoperative Inflammation Markers on the Long-Term Outcomes in Peritoneal Carcinomatosis from Ovarian Cancer"

_cancers, 2024, doi:10.3390/cancers16020254_

Round 1

Reviewer 1 Report

Comments and Suggestions for Authors

The authors performed a study on preoperative inflammatory markers in cases of peritoneal carcinomatosis from epithelial ovarian cancer (OC).

The authors carefully studied the problem by excluding a series of confounding factors such as the presence of immunological or hematological comorbidities, parameters that can affect inflammatory markers (a limitation of these biomarkers).

I have some requests for the authors, however.

  • specify what you mean by “other comorbidities”

  • specify what are the 5 histotypes “others”

  • is the BRCA or HRD data available? If so, I would include it in the paper.

  • in case of multiple comparisons (not paired 2X2) have the authors thought of using statistical correctors to identify specific subgroup differences? (e.g. Bonferroni correction?)

  • I would consider for the discussion some papers on the role of these biomarkers in OC that have not been considered:

  • 10.1038/s41598-021-90361-w
  • 10.1186/s12885-018-4318-5
  • 10.1038/s41598-020-75316-x
  • recently some authors have observed a prognostic role for these markers also for early-stage OC consider adding a brief comment on this issue considering these studies:

  • https://doi.org/10.1002/ijgo.15014
  • 10.1007/s10147-022-02272-z 
  • I would suggest moving the limitations to the end of the discussion, not as the last message in the conclusions (not very appropriate) and I would also add as a limitation the inclusion of non-serous histotypes that have a different biological behavior. I would also emphasize the strengths of the study (in short, I would make a classic section of strengths and limitations in the discussion).
Comments on the Quality of English Language

Minor check 

Author Response

Thank you for your time to review our paper. We responded to all your demands excepting the one regarding the HRD status due to the fact that it is not routinely available in Romania. Please find attached the revised manuscript

Reviewer 2 Report

Comments and Suggestions for Authors

While reading the article, I had some comments and recommendations:

1. The abstract does not understand the design of the study. The authors indicate that 57 patients were included in the retrospective study (their diagnosis was not indicated). Then the Authors immediately write (lines 50-57) that the multivariate analysis prospectively showed that “..In multivariate analysis MLR and SII were significantly associated with higher values of overall survival (p<0.0001 and p=0.0124); meanwhile, preoperative values of CA125, PLR and MLR were not associated with the values of overall survival (p=0.5612, p=0.6137 and p=0.1982 perspectively).”

To improve the presentation of the article, it is necessary to change the abstract.

1.       Section 2 . Materials and Methods is written very briefly.

There is no information on the clinicopathological characteristics of patients. I recommend adding data in the form of a table, which will indicate age, TNM stage, histological type of tumor, number of patients High-grade, low-grade, etc.

The authors do not indicate how the general blood test was performed. What hematology analyzer was used?

What method was used to evaluate the CA125 marker? If this is an ELISA method, then the company of the reagents must be indicated.

It is not specified how the SII indicator is calculated? What are the units of measurement?

2.       The phrase on lines 136-139 is not clear. What do the authors mean by the 841000 level for SII?

3.        In the results section on lines 284-287 “..In multivariate analysis MLR and SII were significantly associated with higher values of overall survival (p<0.0001 and p=0.0124); meanwhile, preoperative values of CA125, PLR and MLR were not associated with the values of overall survival (p=0.5612, p=0.6137 and p=0.1982 perspectively).” It is not clear whether MLR is associated or not associated with overall survival?

Author Response

Thank you for taking your time to review our manuscript. We tried to respond to your demands. Please find attached the revised manuscript
